# DeepKnuckle: Deep Learning for Finger Knuckle Print Recognition

**Ahmad S. Tarawneh [1], Ahmad B. Hassanat [2], Esra'a Alkafaween [2], Bayan Sarayrah [2], Sami Mnasri [3], Ghada A. Altarawneh [4], Malek Alrashidi [3], Mansoor Alghamdi [3] and Abdullah Almuhaimeed [5,\*]**

1   Faculty of Informatics, Eotvos Lorand University, 1117 Budapest, Hungary; Ahmadtr@caesar.elte.hu
2   Faculty of Information Technology, Mutah University, Mutah, Karak 6171, Jordan; hasanat@Mutah.edu.jo (A.B.H.); esrakafaween86@gmail.com (E.A.); Bayan.91@yahoo.com (B.S.)
3   Department of Computer Science, Applied College, University of Tabuk, Tabuk 71491, Saudi Arabia; smnasri@ut.edu.sa (S.M.); Mqalrashidi@ut.edu.sa (M.A.); malghamdi@ut.edu.sa (M.A.)
4   Department of Accounting, Mutah University, Mutah, Karak 6171, Jordan; Ghadaa@Mutah.edu.jo
5   The National Centre for Genomics and Bioinformatics, King Abdulaziz City for Science and Technology, Riyadh 11442, Saudi Arabia
\*   Correspondence: muhaimed@kacst.edu.sa

**Abstract:** Biometric technology has received a lot of attention in recent years. One of the most prevalent biometric traits is the finger-knuckle print (FKP). Because the dorsal region of the finger is not exposed to surfaces, FKP would be a dependable and trustworthy biometric. We provide an FKP framework that uses the VGG-19 deep learning model to extract deep features from FKP images in this paper. The deep features are collected from the VGG-19 model's fully connected layer 6 (F6) and fully connected layer 7 (F7). After applying multiple preprocessing steps, such as combining features from different layers and performing dimensionality reduction using principal component analysis (PCA), the extracted deep features are put to the test. The proposed system's performance is assessed using experiments on the Delhi Finger Knuckle Dataset employing a variety of common classifiers. The best identification result was obtained when the Artificial neural network (ANN) classifier was applied to the principal components of the averaged feature vector of F6 and F7 deep features, with 95% of the data variance preserved. The findings also demonstrate the feasibility of employing these deep features in an FKP recognition system.

**Keywords:** biometrics; transfer learning; finger-knuckle print recognition; deep learning; VGG-19; PCA

## 1. Introduction

Automated identification solutions have become critical for security and privacy in today's digitally linked world [1]. Biometrics is a type of person recognition method; it is a security solution that differs from typical authentication and identification techniques such as passwords, ID cards, and PIN codes [2].

Biometrics are techniques that use automated ways to objectively validate a system utilizing biological features. It uses physiological or behavioral aspects of humans as a means of authenticating personal identity [3]. Physiological characteristics include those retrieved from the human body, such as iris [4], faces [5,6], retinas, veins [7], fingerprints [8–11], palm prints, finger knuckle print, and DNA, ECG [12–15], while behavioral characteristics include voice, stride, signature, and keystroke [16–20]. Biometrics traits are commonly employed in systems such as Security in IoT [21–25], e-banking [26–29], cloud security [30–33], access control [34–37], network security systems [38–42], and ID cards [43–47], and other applications related to IoT [48–52].

Hand-based authentication systems are models that recognize fingerprints [53–57], palm prints, hand geometry, hand form, and hand veins [58,59]. For a long time, hand-based systems have been in the limelight [60]. The success of the systems is due to the

trustworthy qualities that give stability, acceptance, simplicity, and robustness [61]. Many corporations, industries, and government agencies now rely on hand-based technology for a variety of purposes, including security [62].

Recently, it was discovered that the finger-knuckle (FKP) [63] is also a biometric element that may be employed in a safe authentication system [64]. It is a hand-based characteristic in which images include information about an individual's finger knuckle lines and textures, which may become a unique anatomical feature and be used to identify a person. These characteristics were proposed and examined in order to overcome some of the limits and shortcomings of earlier hand-based technologies [65], as well as the inconsistencies of the low-cost [2] and small-size imaging equipment used to capture the structure [66]. Local feature extraction is performed on the FKP system, followed by a mixture of local and global feature extractions, geometric feature extraction, and ultimately an enhanced acquisition device that includes both major and minor intrinsic features [11,67–70].

Fingerprint, palm print, hand geometry, and hand vein are examples of biometrics that have been thoroughly investigated [71]. FKP can be regarded as a different biometric identifying technology because of its uniqueness [72]. The following are the FKP's distinct benefits over other biometrics: the surface of FKP is difficult to abrade since individuals normally grasp objects with the inner side of their hands. Because of the non-contact nature of the FKP collection [73], users are more likely to accept it [74]. As such, FKP is regarded as one of the most feasible and effective personal identification technologies in the future [75–79].

The finger knuckle print is a global, one-of-a-kind, and persistent biometric pattern that is utilized for extremely exact personal identification. Recent FKP research has focused on robust feature extraction, contactless/unconstrained acquisition, and fusion techniques [80]. However, the literature includes limited work [81–85].

In order to support corporate, industrial, economic, and social change for competitive advantage of firms and nations, and to improve overall human progress, technological innovation plays an important role in society for satisfying needs, achieving goals, and solving problems of adopters directed to supporting corporate, industrial, economic, and social change [86–90]. Awareness technological and social change requires a thorough understanding of scientific developments and new technological trajectories [91–95].

Biometrics technology as being part of technological innovation [96,97], have received a lot of attention in recent years, especially to current techniques like machine learning and artificial intelligence. As a result of the continual search, contemporary methods known as deep learning were developed [98–100]. These approaches received a lot of interest since they can be used to classify image textures [101–105]. Deep learning is a branch of machine learning research that uses learning algorithms to scan many layers of representation to model complicated relationships in data [106,107]. As a result, high-level traits and ideas have been identified based on the lowest of them. Deep structure refers to the hierarchical structure of features, and most of these models are built on the approach of supervised or unsupervised representation learning. Convolutional architectures are one of the most important tools for deep learning success in image classification. Principal component analysis has recently become one of the most popular deep learning methodologies (PCA) [108]. Deep network training is computationally intensive from the outset and necessitates a large volume of tagged data [109–113].

Feature extraction uses deep learning models (convolution neural networks (CNNs)) strategy, which is employed when there is a lack of training data or resources [54]. It can be done by using a pre-trained model, such as VGG, Inception, SqueezeNet, and ResNet, which have been trained on a large dataset project [114] like ImagNet. VGG CNN is made up of various primary structures, each of which is made up of several linked convolutional layers and full-connected layers. The convolutional kernel has a size of $3 \times 3$, while the input has a size of $224 \times 224 \times 3$. In general, the number of layers is centered at 16~19 [115]. VGGs have lately demonstrated outstanding performance in many

computer vision applications. Many firms, including Adobe, Apple, Facebook, Baidu, Google, IBM, Microsoft, NEC, Netflix, and NVIDIA, have lately employed VGG as one of their deep learning methodologies [80]. As a pre-processing model, (VGG-19 CNN) is utilized. The network depth has been increased as compared to typical convolutional neural networks. It employs an alternating structure of numerous convolutional layers and non-linear activation layers, which outperforms a single convolutional layer. The layer structure may extract image features more effectively, apply Max pooling for down sampling, and alter the linear unit (ReLU) as the activation function, that is, choose the greatest value in the image area as the pooled value of the area. The down sampling layer is primarily used to increase the network's anti-distortion capabilities to the image while keeping the sample's key characteristics and minimizing the number of parameters [116].

This study aims at providing an investigation structure for the use of deep learning in supporting FKP recognition using VGG-19 (f6 and f7); thereof, our study intends to achieve the following objectives:

1. Determine the extent to which deep learning can support FKP recognition using the deep VGG-19 method.
2. Examine the impact of dimensionality reduction on the discriminative power of deep features.
3. Identify the best performing classifier on FKP deep features.
4. Determine the authentication performance when using FKP deep features.

The remainder of the paper is broken into the following sections: The literature review on FKP recognition is presented in Section 2. The methodology used to implement the proposed approach, the dataset used, the performance evaluation measures, and the structure of our experimental model are all shown in Section 3. The experimental model's results are presented and discussed in Section 4. Finally, part 5 depicts the study's findings as well as its future endeavors.

## 2. Related Work

There has been a lot of study done on FKP recognition for both identification and authentication security, whether for IoT or more general security [116–121]. These are only a few instances.

Lalithamani et al. presented new work in the field of biometric authentication system based on the master finger joint pattern—compared to current systems with simple classifiers such as SVM, PCA, and LDA. Given its accuracy, the deep learning method is best suited for this authentication. The Convolution Neural Network (CNN) extracts features and compares them optimally with trained images. CNN is trained by backpropagation algorithm with random gradient descent and minibatch learning with the help of neural network [7].

Hammouche et al. suggested a novel technique for FKP authentication based on phase congruency with a Gabor Filter bank [122]. Furthermore, Zhang et al. proposed a novel computing framework with the goal of implementing a new efficient feature extraction approach for FKP recognition. The authors conducted a thorough examination of three often utilized local features: local orientation, local phase, and phase congruency. In addition, they developed a method for effectively calculating all characteristics utilizing phase congruency [123]. For an FKP identification system, Muthukumar and Kavipriya used the Gabor feature with an SVM classifier [124].

Heidari and Chalechale presented a unique FKP biometric system in which the feature extraction is a mix of the entropy-based pattern histogram (EPH) and a set of statistical texture characteristics (SSTF). The genetic algorithm (GA) was used to find the best characteristics among the retrieved features. This has been tested on PolyU dataset [73]. While, Singh and Kant developed a multimodal biometric system for person authentication based on FKP and iris characteristics, in which the PCA approach was utilized for feature extraction and the Neuro fuzzy neural network (NFNN) classifier was employed in the identification stage [125].

Chlaoua et al. have developed a simple deep learning method known as principal component analysis. In their proposed approach, PCA has been involved to learn two-stages of filter banks then a simple binary hashing and block histograms for clustering of feature vectors are used. The resultant features are finally fed to the classification step, i.e., linear multiclass Support Vector Machine (SVM). The authors also studied a multimodal biometric system based on matching score level fusion [114].

In Reference [126], a simple CNN end-to-end model is introduced for FKP recognition. The dataset Poly-U FKP is used to evaluate the proposed model and the results gives 99.83% (0.76 and 99.18%, respectively) as the best (standard deviation, mean, respectively) accuracy of recognition. Despite this model being based on a straightforward method for data augmentation with a reduced number of trainable parameters, the latter is composed of a few sets of layers: two connected layers and three convolutional ones. Chalabi et al. also used the PCANet-SVM approach to create a system based on score level fusion of minor and major finger knuckles [62].

Hamidi et al. used two types of pre-trained models, VGG–16 and VGG19, with deep convolutional neural networks to extract features from Finger-Knuckle-Print images in order to construct an efficient multimodal identification system. The results presented in this work reveal that unimodal and multimodal identification systems based on matching score level fusion function extremely well [80].

Fei et al. introduced a new direction convolution difference vector to effectively depict the direction details of finger knuckle images. They then presented a feature learning approach for encoded discriminative direction features for finger knuckle image recognition. The final experimental findings indicate that the proposed method outperforms other Finger knuckle image identification algorithms, demonstrating the usefulness of hash learning-based methods [127].

Many deep learning models have been used for different computer vision tasks. For example, ResNet [128] is a robust model that used an interesting idea of adding the identity block to the CNN architecture in order to make the CNN deeper and reduce the problem of learning degradation in deep CNNs.

Likewise, MobileNet [129] is another interesting CNN which used inverted residual in which the residual connections are created between the bottleneck layers. The small size of this model makes it suitable to be used on mobile devices.

Moreover, ShuffleNet [130] is another CNN model which is suitable for mobile devices. The low computational power of this architecture makes it the right option for many applications. The main reason of this low computational cost is due to two operations used in this CNN, which are channel shuffle and pointwise group convolution.

EfficientNet [131] is a state-of-the-art CNN which introduced exciting ideas to improve the performance of CNNs. EfficientNet suggests that the performance of a CNN can be improved by increasing the depth, number of layers, and the width, number of filters, as well as the resolution of the input image. This architecture has many versions, varying from b0 to b7, each of which has its own depth, width, and resolution scales.

FKP is generally fused with other biometrics to reinforce the intended security system in order to increase the performance of a biometric security system; these approaches include, but are not limited to, the work of [73,132–135].

Despite the large number of FKP recognition methods in the literature [136,137], none are perfect, and each has its own limitations, which are primarily caused by issues inherited from computer vision and machine learning. As a result, there is still room for improvement, especially when using deep learning techniques on FKP images.

## 3. The Proposed Method

The proposed study aims at investigating the ability of deep features extracted from VGG-19 to improving the performance of FKP biometric recognition.

### 3.1. Dataset Description

To achieve the objectives of the study, especially as it employs deep learning, we adopt a standard FKP image dataset called IIT Delhi Finger Knuckle Dataset version1.0.20 [138]. The size of this dataset and its diverse FKP images alignment with VGG-19 layered network for deep architecture design. This helps to overcome the limitations associated with acquiescing real-time as this dataset is publicly available for research community. Therefore, our experiment was carried out on IIT Delhi Finger Knuckle Dataset version1.0.20. IIT Delhi Finger Knuckle Dataset contains 790 images for 158 subjects, 5 images each. The size of each image is 80 × 100 pixels. Figure 1 shows sample images of the used dataset.

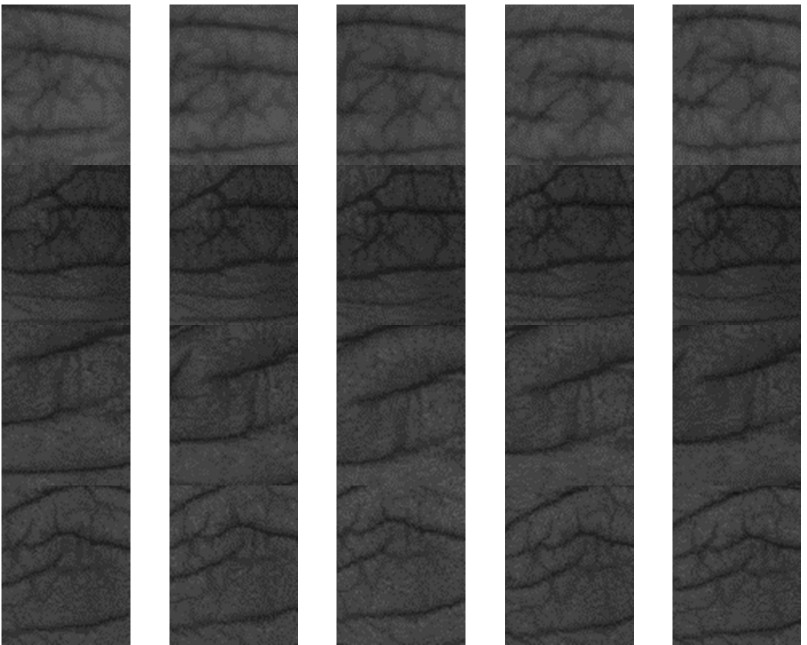

**Figure 1.** Sample from the IIT Delhi Finger Knuckle Dataset [138].

### 3.2. The Proposed FKP Recognition

This section presents the proposed framework and procedures for applying the current investigation to obtain the results of using the VGG-19 deep learning-based method for FKP identification or authentication. Figure 2 shows the proposed framework.

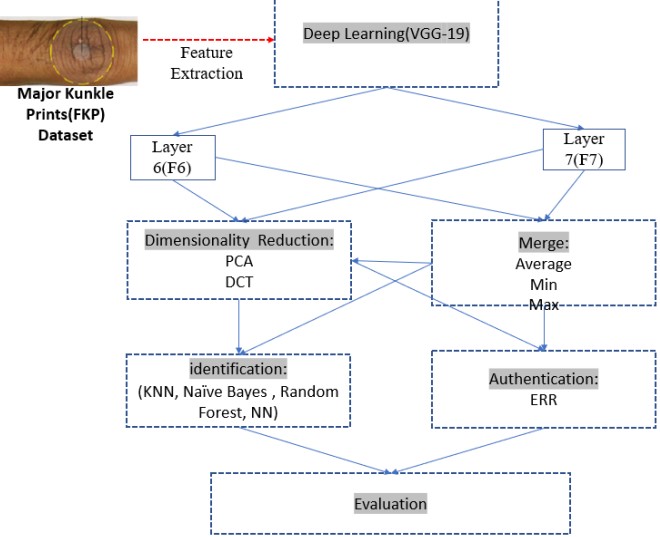

**Figure 2.** The Proposed FKP recognition Model.

In order to extract features, we used the Pre-trained VGG-19 [139] which was pre-trained on the ImageNet dataset [140]. For feature extraction and training, we used MAT-LAB 2017a and WEKA tool. We used the Weka tool version (3.6.9) to train and test using the extracted features, and to classify the obtained deep features using some of its supported classification methods, such as ANN, RF, NB, and KNN. All the WEKA methods are used with their default parameters.

MATLAB relies on data format in matrices; so, we use it to run VGG-19 (layers F6 and F7) for feature extraction, and the dimensionality reduction using PCA, in addition to the computation of average, max and min of the deep features merging (discussed later). The output of this phase is two sets of features vectors:

- F6, which is collected from layer 6.
- F7, which is collected from layer 7.

Each set has 790 rows, which are the number of images and examples in the FKP dataset, and 4096 columns, which are the number of features obtained by the pre-trained VGG-19 for each layer. The resultant feature vectors are used for training and testing purposes. F8, on the other hand is a 1000-node classification layer that represents the likelihood of one class in the ImageNet dataset. It is not a common practice to use it here because it does not contain deep features.

After getting the feature vectors, we converted them to an arff file to be compatible with the WEKA tool [48].

The vectors obtained using the pre-trained VGG-19 are very large, 4096 dimensions, therefore, we used WEKA to reduce the number of features. In PCA, the more principal components (PCs) we keep the more dimensionality we get, and vice versa. Therefore, we opt for including all PCs in a manner that retains 95%, 97%, or 99% variance of the data (the 4096 features). Accordingly, we extracted new features sets: PCA99, PCA97, and PCA95, which are reduced versions of the original features. Following this, we used the following classifiers on the new PCA99, PCA97, and PCA95 datasets:

- ANN (Multilayer Perceptron: Weka default parameters: number of hidden layers = (features + classes)/2, learning rate = 0.3).
- RF (Weka default parameters: number of trees in the random forest = 100).
- NB (Weka default parameters: batch size = 100).
- KNN (Weka default parameters: K = 1, linear search with Euclidean distance).

Table 1 shows the number of features after applying the three variance percentages (99%, 97%, and 95%) of PCA to the feature vector of the FKPS datasets (F6 and F7).

**Table 1.** Number of the features of the training set after the PCA applied.

| PCA | Feature Vector | Number of Features |
|---|---|---|
| Raw features vector | VGG19F6 | 4096 |
| | VGG19F7 | 4096 |
| PCA 99% | VGG19F6 | 450 |
| | VGG19F7 | 184 |
| PCA 97% | VGG19F6 | 271 |
| | VGG19F7 | 95 |
| PCA 95% | VGG19F6 | 190 |
| | VGG19F7 | 68 |

### 3.3. Merging F6 and F7

Due to the varying performance of the F6 and F7 features, as will be presented soon, we opt for creating new feature sets by merging both F6 and F7, wishing to obtain deep feature vectors with an improved performance, the merging is done here using any of three methods:

- Minimum:

The minimum merging work by taking the minimum value from each pair of values from F6 and F7. Algorithm 1 explains the minimum merging procedure.

---

**Algorithm 1** Merging of F6 and F7 using minimum method

---

*Input: Feature vector F6 and Feature vector F7*
*Output: Merged feature vector*
    *1: Merged feature vector = zeros(length(F6))*
    *2: for i = 0 to length(F6))*
    *3:   Merged feature vector$_i$ = min(F6$_i$,F7$_i$)*
    4: end for
    5: return *Merged feature vector*

---

- Maximum

The maximum merging works by taking the maximum value from each pair of values from F6 and F7. Algorithm 2 explains the maximum merging procedure.

---

**Algorithm 2** Merging of F6 and F7 using maximum method

---

*Input: Feature vector F6 and Feature vector F7*
*Output: Merged feature vector*
    *1: Merged feature vector = zeros(length(F6))*
    *2: for i = 0 to length(F6))*
    *3:   Merged feature vector$_i$ = max(F6$_i$,F7$_i$)*
    4: end for
    5: return *Merged feature vector*

---

- Average of both F6 and F7:

The average merging works by taking the average value from each pair of values from F6 and F7. Algorithm 3 explains the average merging procedure.

---

**Algorithm 3** Merging of F6 and F7 using average method

---

*Input: Feature vector F6 and Feature vector F7*
*Output: Merged feature vector*
    *1: Merged feature vector = zeros(length(F6))*
    *2: for i = 0 to length(F6))*
    *3:   Merged feature vector$_i$ = mean(F6$_i$,F7$_i$)*
    4: end for
    5: return *Merged feature vector*

---

The three previously mentioned algorithms loop through two feature vectors, F6 and F7. Then each value in the new merged vector is obtained by taking the minimum, maximum, or average of each pair of values. By applying Algorithms 1–3, we get three new deep feature sets; again, we used the dimensionality reduction on these datasets using both PCA. Table 2 shows the number of features after applying PCA with the three variances of data percentages (99%, 97%, and 95%) on the merged feature vectors (F6 and F7), according to average, minimum, and maximum. We call these new deep feature sets MPCA99%, MPCA97%, and MPCA95%.

### 3.4. Identification and Authentication

The evaluation process of this type of model is done through many different performance measures such as Accuracy, Precision, Recall, and F-score, the values of these measures ranging from 0 to 1, 0 represents the worst performance and 1 represents the best performance. We used 10-Fold cross-validation to verify the reliability of the entered data and validate each method and its recognition ability. In each fold, 79 out of 790 images were tested and 711 images used for training, we mean by "image", the resultant deep feature vector that represents each image.

**Table 2.** Number of features before and after applying PCA on the merged F6⊕F7.

|  | Merge Rule | Number of Features |
|---|---|---|
| Raw features | Min | 4096 |
|  | Max | 4096 |
|  | Average | 4096 |
| MPCA 99% | Min | 557 |
|  | Max | 583 |
|  | Average | 431 |
| MPCA 97% | Min | 370 |
|  | Max | 402 |
|  | Average | 250 |
| MPCA 95% | Min | 269 |
|  | Max | 299 |
|  | Average | 173 |

For authentication, five random profiles were chosen, each with five FKP images and a class 1 label. The remaining cases were labeled as class 0. We compared the first profile to all instances in class 0, then profile 2, and so on for all five profiles, using WEKA software. Here we used 5-fold cross validation because we had only 5 images for each subject [141].

## 4. Results and Discussion

In this section, we present and discuss the experimental results of the proposed framework.

### 4.1. Experiments Settings

The presented results show the performance of machine learning classifiers on 20 different feature datasets created from the obtained different deep features of the IIT DELHI image dataset. The feature datasets created are listed as follows:

1. The pure F6 feature vector with 4096 dimensions.
2. The pure F7 feature vector with 4096 dimensions.
3. F6 after applying PCA 95%.
4. F6 after applying PCA 97%.
5. F6 after applying PCA 99%.
6. F7 after applying 95%.
7. F7 after applying PCA 97%.
8. F7 after applying PCA 99%.
9. Average feature vector (F6⊕F7).
10. Minimum feature vector (F6⊕F7).
11. Maximum feature vector (F6⊕F7).
12. Average feature vector (F6⊕F7) after applying PCA 95%.
13. Average feature vector (F6⊕F7) after applying PCA 97%.
14. Average feature vector (F6⊕F7) after applying PCA 99%.
15. Minimum feature vector (F6⊕F7) after applying PCA 95%.
16. Minimum feature vector (F6⊕F7) after applying PCA 97%.
17. Minimum feature vector (F6⊕F7) after applying PCA 99%.
18. Maximum feature vector (F6⊕F7) after applying PCA 95%.
19. Maximum feature vector (F6⊕F7) after applying PCA 97%.
20. Maximum feature vector (F6⊕F7) after applying PCA 99%.

The following tables demonstrate the performance metrics for different classifiers while using the above-designed feature vectors.

As shown in Table 3, the identification results of applying VGG-19(F6) on the IIT DELHI dataset show that the KNN achieves the highest accuracy with (81.5%); followed by the RF with (73.4%). While the lowest accuracy was achieved by NB with (68.6%). As for time, KNN spends less time to conduct the training and testing phases with (0.02 s);

while RF spends the maximum time to accomplish the task of predicting with (1.8 s), the training time of the KNN classifier is always zero because it has no training model, each test example is compared directly to the other examples in the training set, this is one reason why KNN is slow in testing, particularly when we have a large number of examples in the training set [142–145].

**Table 3.** Identification results of VGG-19-F6 deep features. Bold values signify the best performance.

| Classifier | Accuracy | Precision | Recall | F-Measure | Training Time (seconds) |
|---|---|---|---|---|---|
| KNN | **0.815** | **0.815** | **0.814** | **0.814** | 0 |
| RF | 0.734 | 0.725 | 0.735 | 0.730 | 1.8 |
| NB | 0.686 | 0.686 | 0.717 | 0.701 | 1.2 |

As shown in Table 4, the identification results of applying VGG-19(F7) on the IIT DELHI dataset show that the KNN achieves the highest accuracy with (72.8%); followed by the RF with (69.2%). While the lowest accuracy was achieved by NB with (60.08%). For both F6 and F7, the KNN with acceptable accuracy outperformed all other classifiers. Figure 3 illustrates the accuracy of identification using VGG-19-F6 and VGG-19-F7 deep features, showing that F6 is better for identification regardless of the classifier employed.

**Table 4.** Identification results of VGG-19-F7 deep features. Bold values signify the best performance.

| Classifier | Accuracy | Precision | Recall | F-Measure | Training Time (seconds) |
|---|---|---|---|---|---|
| KNN | **0.728** | **0.727** | **0.736** | **0.731** | 0 |
| RF | 0.692 | 0.692 | 0.679 | 0.685 | 1.52 |
| NB | 0.608 | 0.646 | 0.666 | 0.656 | 1.16 |

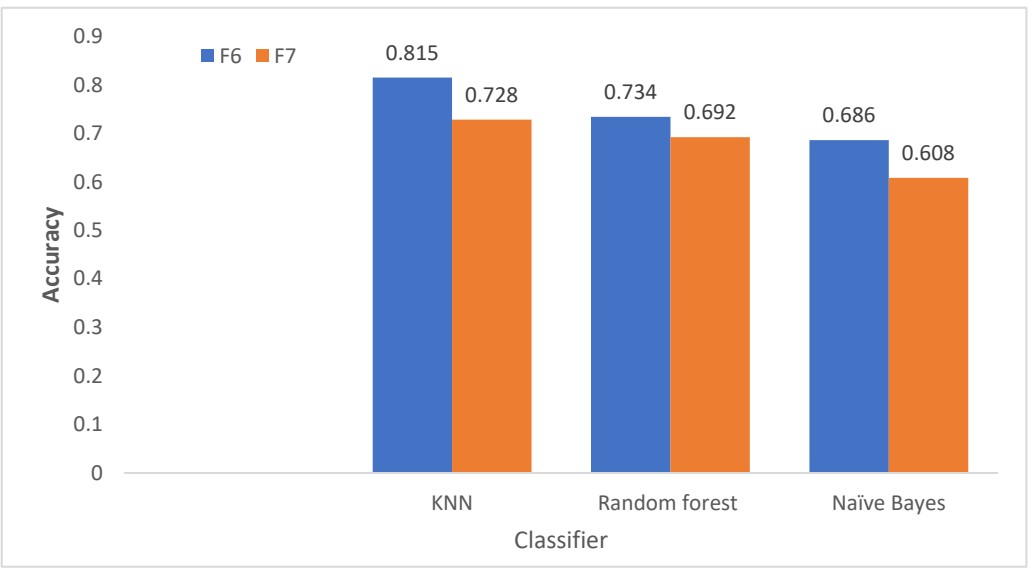

**Figure 3.** Classifier identification accuracy on each of F6 and F7 deep features. Accuracy results obtained from Tables 3 and 4.

### 4.2. Results of PCA Features

When PCA is applied to the F6 feature vectors, we get a new set of feature vectors with fewer dimensions. The number of features is determined by the proportion of data variance maintained, the higher the data variance, the more features we use. The new dataset is used then as an input to the classification process.

Table 5 shows the results of the PCA on F6 in terms of identification. We utilize PCA to minimize the feature vector while preserving the highest data variance (95%, 97%, and 99%).

The results of different principal components in Table 5 demonstrate that all classifiers performance have significantly improved in terms of time.

**Table 5.** Identification results of VGG-19-F6 deep features with PCA (95%, 97%, and 99%). Bold values signify the best performance.

| Classifier | | Accuracy | Precision | Recall | F-Measure | Training Time (seconds) |
|---|---|---|---|---|---|---|
| PCA 95% | ANN | **0.900** | **0.900** | **0.900** | **0.900** | 474.07 |
| | KNN | 0.686 | 0.741 | 0.766 | 0.753 | 0 |
| | NB | 0.262 | 0.382 | 0.443 | 0.410 | **0.09** |
| | RF | 0.425 | 0.475 | 0.490 | 0.483 | 1.45 |
| PCA 97% | ANN | **0.886** | **0.886** | **0.885** | **0.885** | 410.14 |
| | KNN | 0.530 | 0.671 | 0.708 | 0.689 | 0 |
| | NB | 0.276 | 0.404 | 0.460 | 0.431 | **0.05** |
| | RF | 0.624 | 0.631 | 0.647 | 0.639 | 1.72 |
| PCA 99% | ANN | **0.781** | **0.786** | **0.794** | **0.790** | 422.58 |
| | KNN | 0.512 | 0.529 | 0.587 | 0.557 | 0 |
| | NB | 0.013 | 0.026 | 0.080 | 0.039 | **0.03** |
| | RF | 0.495 | 0.485 | 0.499 | 0.492 | 0.99 |

Table 5 shows that all classifiers achieve low training time, especially for RF and NB; this is due to the dimensionality reduction achieved by the PCA. The highest accuracy resultant from applying PCA of all percentage of data variance used (95%, 97%, and 99%) is in favor of ANN with (90%, 88.6%, and 78.1%) respectively, which can be attributed to the reduced feature vector by the PCA, which removed redundant features since keeping at least 95% of data variance. However, ANN consumed the highest training time in this experiment, it is worth mentioning that we could not use the ANN classifier with raw features because of its unacceptable training time (days), this is another advantage of dimensionality reduction since it allows for faster training and hence more classifiers to be used.

Similarly, we applied PCA of 95%, 97%, 99% on F7 as well, Table 6 shows the identification results.

**Table 6.** Identification results of VGG-19-F7 deep features with PCA (95%, 97%, and 99%). Bold values signify the best performance.

| Classifier | | Accuracy | Precision | Recall | F Measures | Training Time (seconds) |
|---|---|---|---|---|---|---|
| PCA 95% | ANN | **0.849** | **0.848** | **0.853** | **0.850** | 205.33 |
| | KNN | 0.690 | 0.608 | 0.613 | 0.611 | 0 |
| | NB | 0.442 | 0.552 | 0.575 | 0.564 | **0.03** |
| | RF | 0.633 | 0.629 | 0.638 | 0.634 | 0.92 |
| PCA 97% | ANN | **0.873** | **0.873** | **0.872** | **0.872** | 200.39 |
| | KNN | 0.714 | 0.714 | 0.768 | 0.740 | 0 |
| | NB | 0.410 | 0.499 | 0.536 | 0.517 | **0.02** |
| | RF | 0.629 | 0.619 | 0.628 | 0.623 | 0.84 |
| PCA99% | ANN | **0.894** | **0.894** | **0.892** | **0.893** | 193.26 |
| | KNN | 0.677 | 0.677 | 0.759 | 0.716 | 0 |
| | NB | 0.275 | 0.275 | 0.437 | 0.337 | **0.02** |
| | RF | 0.559 | 0.542 | 0.553 | 0.548 | 0.84 |

Table 6 shows that all classifiers improved in terms of training time, if compared to the F6, which is due to the reduced number of features acquired after applying PCA on layer F7 compared to layer F6 (see Table 1), which may be justified by the nature of the deep features collected at layer F7 of the VGG-19. Moreover, the performance of some classifiers, namely NB and RF have improved in terms of accuracy in most cases. Again,

the highest accuracy resultant from applying PCA of (95%, 97%, and 99%) are in favor of ANN with (84.9%, 87.3%, and 89.4%) respectively. However, ANN consumes more training time compared to the other classifiers.

In summary, it can be said that KNN achieves the best results given all performances measures except these of ANN. ANN significantly outperforms all the other classifiers in terms of Accuracy, Precision, Recall, and F-measures.

### 4.3. Results after Merging F6 and F7

Both deep features (F6, F7) are merged by setting the average, maximum, and minimum as merging rules to evaluate the identification of the FKP using KNN, NB, and RF classifiers. Table 7 shows the identification results.

**Table 7.** Identification results of VGG-19-deep features obtained by averaging (F6 and F7). Bold values signify the best performance.

| Classifier | Accuracy | Precision | Recall | F-Measure | Training Time (seconds) |
|---|---|---|---|---|---|
| KNN | **0.806** | **0.806** | **0.803** | **0.805** | 0 |
| NB | 0.690 | 0.690 | 0.720 | 0.705 | 0.95 |
| RF | 0.729 | 0.729 | 0.717 | 0.723 | 1.44 |

Table 7 shows the identification results of the merged dataset based on averaging F6 and F7, in which KNN has the highest accuracy (80.6%), followed by RF and then by the NB with the lowest accuracy.

Table 8 shows the identification result based on maximizing F6 and F7, where KNN has achieved the highest accuracy of (80.3%). On the other hand, the NB has obtained a low accuracy of (67.7%), and the highest training running time is recorded by the RF (1.4 s).

**Table 8.** Identification results of VGG-19-deep features obtained by maximizing (F6 and F7). Bold values signify the best performance.

| Classifier | Accuracy | Precision | Recall | F-Measure | Training Time (seconds) |
|---|---|---|---|---|---|
| KNN | **0.803** | **0.803** | **0.800** | **0.801** | 0.0 |
| NB | 0.677 | 0.677 | 0.716 | 0.696 | 1.14 |
| RF | 0.709 | 0.718 | 0.725 | 0.721 | 1.4 |

In Table 9, we can see the identification results based on minimizing F6 and F7, where KNN obtained the highest accuracy (77.8%). On the other hand, the NB obtained the lowest accuracy with (64.7%), and the RF performance was in between.

**Table 9.** Identification results of VGG-19-deep features obtained by minimizing (F6 and F7). Bold values signify the best performance.

| Classifier | Accuracy | Precision | Recall | F-Measure | Training Time (seconds) |
|---|---|---|---|---|---|
| KNN | **0.778** | **0.778** | **0.773** | **0.775** | 0.0 |
| NB | 0.647 | 0.691 | 0.710 | 0.700 | 0.94 |
| RF | 0.697 | 0.691 | 0.699 | 0.695 | 1.4 |

Afterwards, we applied the PCA with (95%, 97%, and 99%) on the merged dataset for each resultant feature vectors, Maximum, Minimum, and Average. The results are shown in Tables 10–12.

As can be seen in Table 10, the ANN classifier achieved the highest accuracy compared to the other classifiers; however, it takes a longer training time.

Similar to the maximum deep features, the minimum deep features show that the ANN achieved the highest accuracy compared to the other classifiers; however, it takes a longer training time as shown in Table 11.

**Table 10.** Identification results of VGG-19-deep features obtained by maximizing (F6 and F7) after applying PCA. Bold values signify the best performance.

|  | Classifier | Accuracy | Precision | Recall | F Measure | Training Time (seconds) |
|---|---|---|---|---|---|---|
| (PCA 95%) | ANN | **0.894** | **0.894** | **0.892** | **0.893** | 781.2 |
|  | KNN | 0.624 | 0.715 | 0.746 | 0.730 | 0 |
|  | NB | 0.287 | 0.410 | 0.466 | 0.436 | 1.14 |
|  | RF | 0.657 | 0.645 | 0.660 | 0.652 | **0.14** |
| (PCA 97%) | ANN | **0.846** | **0.846** | **0.839** | **0.842** | 1598.19 |
|  | KNN | 0.457 | 0.648 | 0.690 | 0.669 | 0 |
|  | NB | 0.216 | 0.343 | 0.400 | 0.369 | **0.17** |
|  | RF | 0.567 | 0.590 | 0.602 | 0.596 | 1.31 |
| (PCA 99%) | ANN | **0.592** | **0.565** | **0.579** | **0.572** | 2249.24 |
|  | KNN | 0.176 | 0.505 | 0.565 | 0.533 | 0 |
|  | NB | 0.109 | 0.209 | 0.247 | 0.226 | **0.14** |
|  | RF | 0.466 | 0.439 | 0.450 | 0.444 | 1.63 |

**Table 11.** Identification results of VGG-19-deep features obtained by minimizing (F6 and F7) after applying PCA. Bold values signify the best performance.

|  | Classifier | Accuracy | Precision | Recall | F Measure | Training Time (seconds) |
|---|---|---|---|---|---|---|
| (PCA 95%) | ANN | **0.889** | **0.889** | **0.885** | **0.887** | 695.57 |
|  | KNN | 0.653 | 0.727 | 0.754 | 0.740 | 0 |
|  | NB | 0.301 | 0.423 | 0.480 | 0.450 | **0.05** |
|  | RF | 0.651 | 0.656 | 0.666 | 0.661 | 1.08 |
| (PCA 97%) | ANN | **0.819** | **0.840** | **0.851** | **0.846** | 1054.17 |
|  | KNN | 0.480 | 0.664 | 0.701 | 0.682 | 0 |
|  | NB | 0.220 | 0.346 | 0.405 | 0.373 | **0.08** |
|  | RF | 0.586 | 0.586 | 0.570 | 0.578 | 1.09 |
| (PCA 99%) | ANN | **0.547** | **0.508** | **0.516** | **0.512** | 2118.81 |
|  | KNN | 0.196 | 0.546 | 0.604 | 0.574 | 0 |
|  | NB | 0.157 | 0.253 | 0.299 | 0.274 | **0.14** |
|  | RF | 0.481 | 0.462 | 0.475 | 0.469 | 1.4 |

**Table 12.** Identification results of VGG-19-deep features obtained by averaging (F6 and F7) after applying PCA. Bold values signify the best performance.

|  | Classifier | Accuracy | Precision | Recall | F Measure | Training Time (seconds) |
|---|---|---|---|---|---|---|
| (PCA 95%) | ANN | **0.918** | **0.918** | **0.917** | **0.917** | 772.1 |
|  | KNN | 0.722 | 0.786 | 0.805 | 0.795 | 0 |
|  | NB | 0.384 | 0.485 | 0.527 | 0.505 | **0.12** |
|  | RF | 0.701 | 0.683 | 0.695 | 0.689 | 1.22 |
| (PCA 97%) | ANN | **0.908** | **0.908** | **0.907** | **0.907** | 622.02 |
|  | KNN | 0.594 | 0.686 | 0.721 | 0.703 | 0 |
|  | NB | 0.277 | 0.396 | 0.449 | 0.421 | **0.06** |
|  | RF | 0.622 | 0.603 | 0.616 | 0.609 | 1.04 |
| (PCA 99%) | ANN | **0.822** | **0.838** | **0.847** | **0.842** | 1368.06 |
|  | KNN | 0.296 | 0.568 | 0.622 | 0.594 | 0 |
|  | NB | 0.128 | 0.233 | 0.276 | 0.252 | **0.4** |
|  | RF | 0.463 | 0.451 | 0.464 | 0.458 | 1.85 |

Again, as stated in Table 12, ANN achieved the highest accuracy level among other classifiers; however, it takes longer training time. However, it is worth noting that the ANN and all the classifiers used are in favor of lower data variance (PCA 95%), particularly when

using F6 or merging with F6, perhaps this is due to losing more redundant features. One exception is when using the PCA on F7, where we notice improvement in the classifiers' performance, less redundant data on F7 is perhaps the reason. Table 13 summarizes all of the identification accuracy results.

**Table 13.** Identification accuracy results summery. Bold values signify the best performance.

| Feature Vector | Nuber of Features | ANN | KNN | RF | NB | Reduction Percentage |
|---|---|---|---|---|---|---|
| F6 feature vector with 4096 dimensions. | 4096 | - | **0.815** | **0.734** | 0.686 | 0% |
| F7 with F6 feature vector with 4096 dimensions | 4096 | - | 0.728 | 0.692 | 0.608 | 0% |
| F6 after applying PCA 95%. | 190 | 0.900 | 0.686 | 0.425 | 0.262 | 95% |
| F6 after applying PCA 97%. | 271 | 0.886 | 0.530 | 0.624 | 0.276 | 93% |
| F6 after applying PCA 99%. | 450 | 0.781 | 0.512 | 0.495 | 0.013 | 89% |
| F7 after applying PCA 95%. | **68** | 0.849 | 0.690 | 0.633 | 0.442 | 98% |
| F7after applying PCA 97%. | 95 | 0.873 | 0.714 | 0.619 | 0.499 | 98% |
| F7 after applying PCA 99%. | 184 | 0.894 | 0.677 | 0.559 | 0.275 | 96% |
| Average feature vector (F6 + F7). | 4096 | - | 0.806 | 0.729 | **0.690** | 0% |
| Minimum feature vector (F6 + F7) | 4096 | - | 0.778 | 0.697 | 0.647 | 0% |
| Maximum feature vector (F6 + F7). | 4096 | - | 0.803 | 0.709 | 0.677 | 0% |
| Average (F6 + F7) PCA 95%. | 173 | **0.918** | 0.722 | 0.701 | 0.384 | 96% |
| Average (F6 + F7) PCA 97%. | 250 | 0.908 | 0.594 | 0.622 | 0.277 | 94% |
| Average feature (F6 + F7) PCA 99%. | 431 | 0.822 | 0.296 | 0.463 | 0.128 | 89% |
| Minimum (F6 + F7) PCA 95%. | 269 | 0.889 | 0.727 | 0.656 | 0.423 | 93% |
| Minimum (F6 + F7) PCA 97%. | 370 | 0.819 | 0.480 | 0.586 | 0.220 | 91% |
| Minimum (F6 + F7) PCA 99%. | 557 | 0.547 | 0.196 | 0.481 | 0.157 | 86% |
| Maximum (F6 + F7) PCA 95%. | 299 | 0.894 | 0.624 | 0.657 | 0.287 | 93% |
| Maximum (F6 + F7) PCA 97%. | 402 | 0.846 | 0.457 | 0.567 | 0.216 | 90% |
| Maximum (F6 + F7) PCA 99%. | 583 | 0.592 | 0.176 | 0.466 | 0.109 | 86% |

Table 14 shows a direct comparison of the proposed method to some of the other methods that used the same database, which include Surrounded Patterns Code (SPC) [146], Enhanced Local Line Binary Pattern (ELLBP) [147], Local Binary Patterns (LBP) [148], Centralized Binary Patterns (CBP) [149], Center-Symmetric Local Binary Pattern (CSLBP) [150], and Local Binary Patterns for Finger Outer Knuckle (LBP-FOK) [151]. Moreover, we compared our method to some of the-state-of-the-art deep learning-based feature extraction methods such as ResNet18 [128], MobileNetV2 [129], ShuffleNet [111], and EfficientNetb0 [131]. To extract the deep features using these deep learning-based methods, we employed their pre-trained versions. The features are retrieved from the layer immediately preceding the classification layer. The features are then classified using the ANN after PCA 95% is applied.

**Table 14.** Direct comparison of the proposed method to other methods. Bold values signify the best performance.

| Method | Accuracy (%) |
|---|---|
| Proposed | **91.8** |
| SPC | 54.1 |
| ELLBP | 70.47 |
| LBP | 71.63 |
| CBP | 76.67 |
| CSLBP | 76.74 |
| LBP-FOK | 85.97 |
| Resnet18 | 88.48 |
| MobileNetV2 | 44.85 |
| ShuffleNet | 52.28 |
| EfficientNetb0 | **92.91** |

As presented in Table 14, the proposed method achieves superior results to six hand-crafted methods which are SPC, ELLBP, LBP, CBP, CSLBP, and LBP-FOK. In addition, other deep learning approaches outperformed all hand-crafted methods. Furthermore, the proposed method outperforms MobileNet, ShuffleNet, and Resnet18 when compared to other deep learning methods. EfficientNet, on the other hand, achieved slightly better 1% results than our proposed method.

### 4.4. Authentication

Knowing that Authentication is different from Identification in biometric systems, in this study, to conduct an authentication experiment, five profiles were randomly chosen, each of which has five FKP images labelled as class 1, belonging to one of the subjects. The rest of the images belonging to the rest of the subjects are labelled as class 0. The resultant image dataset is called profile 1, which is used to authenticate subject 1 using the same approach, for the sake of simplicity, we created five profiles only, each authenticate a specific subject. We then obtained the deep features using the same aforementioned methodology, that is we used F6 and F7 features after applying PCA 95%. For the performance assessment, the 5-fold cross-validation was applied with KNN to obtain the averaged authentication results. Note that the main aim here is to find out the ability of these features to give a unique representation of each subject in the FKP images dataset, and therefore, it can be reliably employed for authentication purposes.

For evaluation purpose, the Equal Error Rate (EER) is considered as a measure for typical biometric systems, it is normally used to predetermine the threshold values based on the false acceptance rate and the false rejection rate in a particular biometric system [152]. Tables 15 and 16 present the Authentication EER based on the data profiles created for the purpose of authentication in this study.

**Table 15.** Authentication ERR of VGG-19-F6 deep features after applying PCA (95%).

| Dataset | EER % |
|---|---|
| Profile 1 | 0.5 |
| Profile 2 | 0.12 |
| Profile 3 | 0.12 |
| Profile 4 | 0.12 |
| Profile 5 | 0.37 |
| Avg. | 0.24 |

**Table 16.** Authentication ERR of VGG-19-F7 deep features after applying PCA (95%).

| Experiments | EER % |
|---|---|
| Profile 1 | 0.1 |
| Profile 2 | 0.12 |
| Profile 3 | 0.12 |
| Profile 4 | 0.63 |
| Profile 5 | 0.2 |
| Avg. | 0.23 |

As can be seen from Tables 15 and 16, the average authentication for EER is very small (0.24% when using F6-PCA, and 0.23% when using F7-PCA), having such a small error rate means that the accuracy of the authentication system is very high, this is expected since the number of classes in authentication systems is typically two, while in the identification problem it is equal to the number of subjects in the dataset.

Based on the results in Table 17, our proposed approach, which investigates the use of deep learning for FKP authentication, has a low error rate and a high false rejection rate when compared to the other methods such as the work of [153]. However, to the best of our knowledge, very few previous studies tackled the feasibility of deep learning (VGG-19) in the improvement of FKP recognition using ANN, KNN, NB, or RF. However, several studies discussed the feasibility of FKP recognition using machine learning classifiers.

However, as a matter of fact, the lower EER, the more accurate the process is. Therefore, compared to [153]; our proposed approach maintains a lower EER rate in both cases of F6 and F7 using PCA. Similarly, lower EER when compared with EER by [154].

**Table 17.** Authentication EER of [153] compared to ours. Bold values signify the best performance.

| Methodology | EER % |
|---|---|
| VGG19 F6 + PCA | **0.24** |
| VGG19 F7 + PCA | **0.23** |
| [153] | 1.02 |

*4.5. Discussion*

For the identification/authentication of FKP, our Investigation, which is based on the evidence in hand (the dataset used, and the method used) concludes the following:

- The ANN is the best classifier to be used for deep features extracted using VGG-19, if it is provided with the reduced version of the features, otherwise, i.e., if it is applied on the original pure deep features, which obtained from the VGG-19 layer 6 or 7 or any merging of them both, the training time would be unacceptably long.
- Using the PCA does not only reduce the dimensionality, and therefore, the training time significantly, but also allows for increasing the identification accuracy, particularly when using the ANN classifier.
- In general, the original pure F6 provides more distinctive features compared to F7, this is evident from the first 8 rows in Table 13.
- Merging the deep features from layers 6 and 7 allows for more distinctive features, particularly, when using the averaging rule.
- The dimensionality reduction using PCA on the averaged deep features is the best combination to provide the most distinctive features compared to the other approaches investigated.
- The extracted features from the deep features using PCA with 95% of data variance are more distinctive and smaller in number if compared to the other two percentages (97% and 99%) used, this is evident from the highest accuracy achieved by the ANN classifier on the deep features obtained by averaging (F6+F7) after applying PCA 95%, see Table 13, row 14.
- As for the authentication of FKP our investigation concludes that using deep features obtained from either layer 6 or layer 7 leads to a reasonable FKP authentication system, particularly when applying PCA on these deep features.

**5. Conclusions**

In this paper, deep learning network VGG-19 is investigated to be used for FKP identification or authentication system, using the deep features collected at layers 6 and 7, with and without dimensionality reduction tools such as the PCA, merging both deep features (F6 and F7) is also investigated using different rules such as average, maximum, and minimum.

For the identification system, several machine learning classifiers were applied (such as ANN, KNN, NB, and RF) on deep features obtained from the open-access dataset IIT Delhi Finger Knuckle Dataset. The experimental results show that the best identification result was obtained when applying the ANN classifier on the principal components of the averaged feature vector of F6 and F7 deep features preserving 95% of the data variance. The results also show the potential of using these deep features for an FKP authentication system.

Furthermore, comparisons to state-of-the-art deep learning-based methods demonstrate the superiority of the proposed solution; however, we discovered that another deep learning feature extraction method, EfficientNet, slightly outperforms ours, hence we recommend more research to be conducted on our method and the other deep features method, perhaps incorporating more than one deep features method for FKP recognition.

Aside from the recognition accuracy, the study's limitations include the use of only one FKP dataset, one Deep learning method (VGG), and one dimensionality reduction method (PCA), which we believe is insufficient to make substantial conclusions. As a result, future research will involve using a more standard FKP dataset with a larger number of images and subjects. Moreover, more deep learning networks will be investigated such as AlexNet [155], GoogleNet [156], etc. Besides, other dimensionality reduction techniques can be used for reducing the dimensionality of the deep features such as DCT, wavelet transform, etc., which need to be examined when applied into the deep neural network itself. Moreover, our future work will focus on merging other biometrics with the FKP such as face, fingerprint, and hand shape altogether, to obtain their deep features in order to provide a more accurate biometric system.

**Author Contributions:** All authors have contributed equally to this paper. All authors have read and agreed to the published version of the manuscript.

**Funding:** This research received no external funding.

**Informed Consent Statement:** Not applicable, because the FKP database used is a public and open access image dataset utilized by many researchers. Consent statements obtained by the original authors of the dataset [138].

**Data Availability Statement:** The dataset used in this paper is publicly available at the dataset website: http://www4.comp.polyu.edu.hk/~csajaykr/IITD/iitd_knuckle.htm, accessed on 13 January 2022.

**Conflicts of Interest:** The authors declare no conflict of interest.

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
