# Peer review of "DeepKnuckle: Deep Learning for Finger Knuckle Print Recognition"

_electronics, doi:10.3390/electronics11040513_

Round 1

Reviewer 1 Report

 DeepKnuckle: Deep Learning for Finger Knuckle Print Recognition

The topics of this paper are interesting, but the structure and theoretical framework of this study needs to be improved. This paper needs a major revision by authors before to be reconsidered.

The abstract can be shorter and better describe results with applied implications of this study.

Introduction. Authors have to clarify better the research questions of this study and explain better basic concepts of deep learning and of the evolution of technology based on theories of technological interactions with other technologies contextualized in this domain. In brief, introduction must reinforce innovation studies to support this new technology and role of leading firms  for an industrial change (see suggested readings). 

Methods of this study can be clarified with following three sections, avoiding a lot of subheadings that create fragmentation and confusion; authors, if necessary, can use bullet points:
•    Sample and data
•    Measures of variables 
•    Models, data analysis and computational methods

Methods have also to clarify the hypotheses for technology analysis (see suggested paper).

Many sections can be merged, under three just mentioned. 

Accuracy in y-axis of figure 3 is not clear how it is assessed and must be clarified in the text and in a note to this figure.

Section 4 has also to avoid subheadings that create a fragmentation of the paper, as said, authors can use bullet points. 

Tables,  I suggest avoiding acronyms and write full words or to insert a note to clarify these acronyms, to clarify them in the text is not enough to be clear for readers.  

The text has a lot of tables that are difficult to digest. Authors have to insert the most important ones in the text, the other can be inserted in an appendix. 

 Conclusion has not to be a summary, but authors have to focus on manifold limitations of this study and provide suggestions of technological applications and management of this innovation, similar to other innovations,  and likely future development.

If the paper is improved as suggested that I will in-depth verify, the paper may be considered. 

Suggested readings of papers that must be read, and all inserted in the text and references.

Anand, V., Kanhangad, V.    2022.Cross-Sensor Pore Detection in High-Resolution Fingerprint Images, IEEE Sensors Journal22(1), pp. 555-564

Coccia M. 2021. Technological Innovation. The Blackwell Encyclopedia of Sociology. Edited by George Ritzer and Chris Rojek. John Wiley & Sons, Ltd .DOI: 10.1002/9781405165518.wbeost011.pub2

Lee, S., Jang, S.-W., Kim, D., Hahn, H., Kim, G.-Y.    2021. A Novel Fingerprint Recovery Scheme using Deep Neural Network-based Learning.     Multimedia Tools and Applications 80(26-27), pp. 34121-34135

Coccia M., Finardi U. 2012. Emerging nanotechnological research for future pathways of biomedicine. International Journal of Biomedical nanoscience and nanotechnology, vol. 2, nos. 3-4, pp. 299-317. DOI: 10.1504/IJBNN.2012.051223

Impedovo, D., Dentamaro, V., Abbattista, G., Gattulli, V., Pirlo, G.    2021. A comparative study of shallow learning and deep transfer learning techniques for accurate fingerprints vitality detection. Pattern Recognition Letters 151, pp. 11-18

Coccia M. 2019. A Theory of classification and evolution of technologies within a Generalized Darwinism, Technology Analysis & Strategic Management, vol. 31, n. 5, pp. 517-531, http://dx.doi.org/10.1080/09537325.2018.1523385  

Nogay, H.S.    2021.     Comparative experimental investigation of deep convolutional neural networks for latent fingerprint pattern classification. Traitement du Signal38(5), pp. 1319-1326

Coccia M. 2017. Disruptive firms and industrial change, Journal of Economic and Social Thought, vol. 4, n. 4, pp. 437-450, http://dx.doi.org/10.1453/jest.v4i4.1511

Liu, F., Zhang, W.-T., Liu, H.-Z., Liu, G.-J., Shen, L.-L.    2021Deep Learning Based Fingerprint Subsurface Reconstruction |    Jisuanji Xuebao/Chinese Journal of Computers 44(10), pp. 2033-2046

Coccia M., Finardi U. 2013. New technological trajectories of non-thermal plasma technology in medicine. Int. J. Biomedical Engineering and Technology, vol. 11, n. 4, pp. 337-356, DOI: 10.1504/IJBET.2013.055665

Zhong, C., Xu, P., Zhu, L.    2021    A deep convolutional generative adversarial network-based fake fingerprint generation method    , 2021 IEEE International Conference on Computer Science, Electronic Information Enineering and Intelligent Control Technology, CEI 2021, pp. 63-67

Coccia M. 2020. Multiple working hypotheses for technology analysis, Journal of Economics Bibliography, vol. 7., n. 2, pp. 111-126, http://dx.doi.org/10.1453/jeb.v7i2.2050

Zeng, L., Al-Rifai, M., Nolting, M., Nejd, W.    2021Triplet Loss for Effective Deployment of Deep Learning Based Driver Identification Models        IEEE Conference on Intelligent Transportation Systems, Proceedings, ITSC, 2021-September, pp. 1328-1333

Chawla, B., Tyagi, S., Jain, R., Talegaonkar, A., Srivastava, S.    2021Finger Vein Recognition Using Deep Learning        Advances in Intelligent Systems and Computing, 1164, pp. 69-78

Coccia M. 2017. Sources of technological innovation: Radical and incremental innovation problem-driven to support competitive advantage of firms. Technology Analysis & Strategic Management, vol. 29, n. 9, pp. 1048-1061, https://doi.org/10.1080/09537325.2016.1268682

Author Response

DeepKnuckle: Deep Learning for Finger Knuckle Print Recognition

The topics of this paper are interesting, but the structure and theoretical framework of this study needs to be improved. This paper needs a major revision by authors before to be reconsidered.

Response: Thank you for your time and efforts, yes there was a problem in the structure of our paper, and we have improved it as suggested.

The abstract can be shorter and better describe results with applied implications of this study.

Response: We have improved the abstract as suggested.

Introduction. Authors have to clarify better the research questions of this study and explain better basic concepts of deep learning and of the evolution of technology based on theories of technological interactions with other technologies contextualized in this domain. In brief, introduction must reinforce innovation studies to support this new technology and role of leading firms  for an industrial change (see suggested readings). 

Response: We added all the suggested papers as references and highlighted the research question.

Methods of this study can be clarified with following three sections, avoiding a lot of subheadings that create fragmentation and confusion; authors, if necessary, can use bullet points:
•    Sample and data
•    Measures of variables 
•    Models, data analysis and computational methods

Methods have also to clarify the hypotheses for technology analysis (see suggested paper).

Response: Thank you for the suggested papers, we have learned many things from them to improve our paper.

Many sections can be merged, under three just mentioned. 

Response: We have merged many sections and kept only the suggested three sections.

Accuracy in y-axis of figure 3 is not clear how it is assessed and must be clarified in the text and in a note to this figure.

Response: This is the accuracy of identification using VGG-19-F6 and VGG-19-F7 deep features, we clarified this in the text and the caption of the figure.

Section 4 has also to avoid subheadings that create a fragmentation of the paper, as said, authors can use bullet points. 

Response: Subheadings are avoided now all over the paper.

Tables,  I suggest avoiding acronyms and write full words or to insert a note to clarify these acronyms, to clarify them in the text is not enough to be clear for readers.  

The text has a lot of tables that are difficult to digest. Authors have to insert the most important ones in the text, the other can be inserted in an appendix. 

Response: Yes, inserting these tables in an appendix is a good idea, however, we could not do that because we were investigating to find the best solution, and these tables show the sequence of investigation to reach the conclusion that we are after.

Conclusion has not to be a summary, but authors have to focus on manifold limitations of this study and provide suggestions of technological applications and management of this innovation, similar to other innovations, and likely future development.

Response: Yes, there were problems in the conclusion part, we fixed this section, adding the limitations of the study, and our future work to address them.

If the paper is improved as suggested that I will in-depth verify, the paper may be considered. 

Suggested readings of papers that must be read, and all inserted in the text and references.

Response: Thank you for these interesting papers, we added all of the following papers as valuable references, in addition to some other similar ones, which improved our paper.

Anand, V., Kanhangad, V.    2022.Cross-Sensor Pore Detection in High-Resolution Fingerprint Images, IEEE Sensors Journal22(1), pp. 555-564

Coccia M. 2021. Technological Innovation. The Blackwell Encyclopedia of Sociology. Edited by George Ritzer and Chris Rojek. John Wiley & Sons, Ltd .DOI: 10.1002/9781405165518.wbeost011.pub2

Lee, S., Jang, S.-W., Kim, D., Hahn, H., Kim, G.-Y.    2021. A Novel Fingerprint Recovery Scheme using Deep Neural Network-based Learning.     Multimedia Tools and Applications 80(26-27), pp. 34121-34135

Coccia M., Finardi U. 2012. Emerging nanotechnological research for future pathways of biomedicine. International Journal of Biomedical nanoscience and nanotechnology, vol. 2, nos. 3-4, pp. 299-317. DOI: 10.1504/IJBNN.2012.051223

Impedovo, D., Dentamaro, V., Abbattista, G., Gattulli, V., Pirlo, G.    2021. A comparative study of shallow learning and deep transfer learning techniques for accurate fingerprints vitality detection. Pattern Recognition Letters 151, pp. 11-18

Coccia M. 2019. A Theory of classification and evolution of technologies within a Generalized Darwinism, Technology Analysis & Strategic Management, vol. 31, n. 5, pp. 517-531, http://dx.doi.org/10.1080/09537325.2018.1523385  

Nogay, H.S.    2021.     Comparative experimental investigation of deep convolutional neural networks for latent fingerprint pattern classification. Traitement du Signal38(5), pp. 1319-1326

Coccia M. 2017. Disruptive firms and industrial change, Journal of Economic and Social Thought, vol. 4, n. 4, pp. 437-450, http://dx.doi.org/10.1453/jest.v4i4.1511

Liu, F., Zhang, W.-T., Liu, H.-Z., Liu, G.-J., Shen, L.-L.    2021Deep Learning Based Fingerprint Subsurface Reconstruction |    Jisuanji Xuebao/Chinese Journal of Computers 44(10), pp. 2033-2046

Coccia M., Finardi U. 2013. New technological trajectories of non-thermal plasma technology in medicine. Int. J. Biomedical Engineering and Technology, vol. 11, n. 4, pp. 337-356, DOI: 10.1504/IJBET.2013.055665

Zhong, C., Xu, P., Zhu, L.    2021    A deep convolutional generative adversarial network-based fake fingerprint generation method    , 2021 IEEE International Conference on Computer Science, Electronic Information Enineering and Intelligent Control Technology, CEI 2021, pp. 63-67

Coccia M. 2020. Multiple working hypotheses for technology analysis, Journal of Economics Bibliography, vol. 7., n. 2, pp. 111-126, http://dx.doi.org/10.1453/jeb.v7i2.2050

Zeng, L., Al-Rifai, M., Nolting, M., Nejd, W.    2021Triplet Loss for Effective Deployment of Deep Learning Based Driver Identification Models        IEEE Conference on Intelligent Transportation Systems, Proceedings, ITSC, 2021-September, pp. 1328-1333

Chawla, B., Tyagi, S., Jain, R., Talegaonkar, A., Srivastava, S.    2021Finger Vein Recognition Using Deep Learning        Advances in Intelligent Systems and Computing, 1164, pp. 69-78

Coccia M. 2017. Sources of technological innovation: Radical and incremental innovation problem-driven to support competitive advantage of firms. Technology Analysis & Strategic Management, vol. 29, n. 9, pp. 1048-1061, https://doi.org/10.1080/09537325.2016.1268682

Reviewer 2 Report

The focus of this manuscript is about finger knuckle biometric identity recognition, where a group of the classifiers, including RF, ANN, NB, KNN, were evaluated followed by the features extracted via VGG-19 nets, of course after PCA processing. This manuscript was well organized and written. The designing of the experiments indeed solves the problem by proposed methods, besides, answering the questions of how varying combinations of feature processings affect results, including PCA, min, max, and average.  This paper is valuable to be published.

Line 44-47, no more than five references at once.

Line 214, why choose F6 and F7, but not include F8? Clarifying it.

Author Response

The focus of this manuscript is about finger knuckle biometric identity recognition, where a group of the classifiers, including RF, ANN, NB, KNN, were evaluated followed by the features extracted via VGG-19 nets, of course after PCA processing. This manuscript was well organized and written. The designing of the experiments indeed solves the problem by proposed methods, besides, answering the questions of how varying combinations of feature processings affect results, including PCA, min, max, and average.  This paper is valuable to be published.

Response: Thank you for your time and efforts and for these encouraging comments.

Line 44-47, no more than five references at once.

Response: We fixed this issue.

Line 214, why choose F6 and F7, but not include F8? Clarifying it.

Response: F8 is a 1000-node classification layer that represents the likelihood of one class in the ImageNet dataset. It is not a common practice to use it. More crucially, we used f6 and f7 to combine the deep features of two layers with the same dimensionality to undertake more trials and tests. We clarified this in the text on page 7.

Reviewer 3 Report

This paper studies an interesting topic: knuckle print recognition. It may have wide applications in forensics and online payment.

The references are adequate, and the literature review can be improved by further comparison of the existing works. Some of the latest neural network methods should be further reviewed, such as residual network. Instead of just mentioning it, further comparison should be provided.

Why VGG is chosen instead of other models? In the experiment section, VGG is not futher compared with other neural network architectures. I would suggest to compare it with at least two SOTA neural networks in this field.

The authors studied the feature dimension reduction using several conventional methods, such as PCA. Can we merge this feature reduction into the process of representation learning in deep neural network?

WEKA is used as a classification tool. Please provide detailed settings for each classifier, so that the results can be repeated by interested readers.

Author Response

The references are adequate, and the literature review can be improved by further comparison of the existing works. Some of the latest neural network methods should be further reviewed, such as residual network. Instead of just mentioning it, further comparison should be provided.

Response: Thank you for your time and efforts and for these encouraging comments. We have added four more state-of-the-art studies (in page 5) and more comparisons to these studies have been conducted and added to the results section (on page 16).

Why VGG is chosen instead of other models? In the experiment section, VGG is not futher compared with other neural network architectures. I would suggest to compare it with at least two SOTA neural networks in this field.

Response: Thank you for this comment. We selected VGG model due to its great success in many Computer Vision tasks. Also, we believe that VGG layers structure and kernel size is suitable for the problem of this research. The added results to the manuscript confirm our intuition regarding the selection of VGG model. And we compared to four deep learning methods.

The authors studied the feature dimension reduction using several conventional methods, such as PCA. Can we merge this feature reduction into the process of representation learning in deep neural network?

Response: To the best of our knowledge, PCA is a statistical-based method to project the features from one space, high dimensional space, to another, low dimensional space. Designing a network to perform PCA is doable as neural networks can approximate any given function if the right architecture design is provided. However, doing this needs training on large data and consumes more time than doing it in the simplest known way, like in our case. Additionally, there are deep learning architectures which proved to give results similar to PCA, like auto encoders, but these architectures need to be trained on large datasets. However, this interesting idea might be successful, but need a lot of work and time, we mentioned this idea in future work part.

WEKA is used as a classification tool. Please provide detailed settings for each classifier, so that the results can be repeated by interested readers.

Response: Thank you for this suggestion. In fact, we used all WEKA methods with their default parameters. We added a statement to the manuscript stating this practice, and we mentioned these default features as well on page 7.

Round 2

Reviewer 1 Report

I have read thoroughly the revised version of paper.

Now this version of the paper, after revision done,

is OK and provides interesting results for readers.